# Population Dynamics of Insect Pests and Beneficials on Different Snap Bean Cultivars

**DOI:** 10.3390/insects14030230

**Published:** 2023-02-25

**Authors:** Yinping Li, George N. Mbata, Alvin M. Simmons

**Affiliations:** 1Agricultural Research Station, Fort Valley State University, 1005 State University Drive, Fort Valley, GA 31030, USA; 2U.S. Vegetable Laboratory, U.S. Department of Agriculture-Agricultural Research Service, 2700 Savannah Highway, Charleston, SC 29414, USA

**Keywords:** host plant resistance, temperature, *Bemisia tabaci*, *Phaseolus vulgaris*, cucumber beetle, Mexican bean beetle, thrips, tarnished plant bug, potato leafhopper, pollinators

## Abstract

**Simple Summary:**

A study was conducted to survey the populations of insect pests and beneficials on different cultivars of snap bean in Georgia, USA. It is important to conserve the beneficials and to understand both the abundance and diversity of beneficials and pests in crops. The population dynamics of insect pests, pollinators, and natural enemies were evaluated on 24 snap bean cultivars weekly for six weeks. The number of sweetpotato whitefly eggs was lowest on cultivar ‘Jade’, whereas cultivars ‘Gold Mine’, ‘Golden Rod’, ‘Long Tendergreen’, and ‘Royal Burgundy’ supported the fewest whitefly nymphs. Cultivars ‘Greencrop’ and ‘PV-857′ harbored fewer adult potato leafhoppers and tarnished plant bugs. The population peaks of adults were observed in Week 1 (25 days after plants emerged) for whitefly and Mexican bean beetle; Week 3 for cucumber beetle, kudzu bug, and potato leafhopper; Weeks 3 and 4 for thrips; Week 4 for tarnished plant bugs; and Weeks 5 and 6 for bees. Temperature and relative humidity correlated with the populations of whitefly, Mexican bean beetle, bees, and predator ladybird beetle. These results contribute crucial information to the agricultural community for the management of insect pests on snap beans.

**Abstract:**

Snap bean is an important crop in the United States. Insecticides are commonly used against pests on snap bean, but many pests have developed resistance to the insecticides and beneficials are threatened by the insecticides. Therefore, host plant resistance is a sustainable alternative. Population dynamics of insect pests and beneficials were assessed on 24 snap bean cultivars every week for six weeks. The lowest number of sweetpotato whitefly (*Bemisia tabaci*) eggs was observed on cultivar ‘Jade’, and the fewest nymphs were found on cultivars ‘Gold Mine’, ‘Golden Rod’, ‘Long Tendergreen’, and ‘Royal Burgundy’. The numbers of potato leafhopper (*Empoasca fabae*) and tarnished plant bug (*Lygus lineolaris*) adults were the lowest on cultivars ‘Greencrop’ and ‘PV-857′. The highest numbers of adults were found in Week 1 (25 days following plant emergence) for *B. tabaci* and Mexican bean beetle (*Epilachna varivestis*); Week 3 for cucumber beetle, kudzu bug (*Megacopta cribraria*), and *E. fabae*; Weeks 3 and 4 for thrips; Week 4 for *L. lineolaris*; and Weeks 5 and 6 for bees. Temperature and relative humidity correlated with *B. tabaci*, *E. varivestis*, bee, and predator ladybird beetle populations. These results provide valuable information on the integrated pest management of snap beans.

## 1. Introduction

Snap bean (*Phaseolus vulgaris* L. (Fabales: Fabaceae)) is an economically important vegetable in the United States, including the state of Georgia [1,2]. The production of snap bean for all uses averaged 330,693 tons during 2018–2020 in the United States [2]. The 2017 United States Census of Agriculture reported Florida, Georgia, Tennessee, California, Texas, North Carolina, New Jersey, Ohio, and New York as the leading fresh-market snap bean states, making up about 80% of harvested snap bean for the year [2]. Georgia ranked second in the United States snap bean acreage and accounted for approximately 18% of United States planted acreage in 2000 [1]. In Georgia, snap bean was valued at about $21 million in 2000 [1]. Snap bean ranked second in acreage and sixth in value for Georgian vegetables [1].

Pests are among the foremost threats to snap bean production [3]. The most common crop pests of snap bean in Georgia are defoliators (caterpillars and beetles), leaf-sucking pests (whiteflies and thrips), and pod feeders (corn earworm, *Helicoverpa zea* (Boddie) (Lepidoptera: Noctuidae); stink bugs; and European corn borer, *Ostrinia nubilalis* (H übner) (Lepidoptera: Pyralidae)) [4]. These pests result in extensive economic losses of snap bean [1]. For example, in 2000, pests caused economic losses of approximately $4.4 million (damage + control cost) worth of snap beans [1]. Among those economic losses, corn earworm and other lepidopteran pests were responsible for the most losses ($1.6 million), followed by thrips ($1.4 million), stink bugs ($900,000), and European corn borer ($500,000) [1]. Moreover, whitefly and whitefly-transmitted viral diseases (e.g., *Cucurbit leaf crumple virus* and *Sida golden mosaic virus*) caused a 40% and 45% reduction in snap bean value in 2016 and 2017, respectively [5,6]. Applications of chemical insecticides are among main tools commonly used by vegetable growers to manage pests in vegetables including snap bean [7]. However, many pests on snap bean have developed resistance to most chemical insecticides [8,9]. Thus, alternative management strategies, such as the use of host plant resistance, are warranted to mitigate losses of snap bean by pests.

Host plant resistance, where crops are bred or genetically engineered to reduce pest damage, has become an essential pest management tool [10]. Several snap bean cultivars have demonstrated certain levels of resistance to some pests including the sweetpotato whitefly, *Bemisia tabaci* (Gennadius) (Hemiptera: Aleyrodidae) Middle East-Asia Minor 1 [11,12,13]; potato leafhopper, *Empoasca fabae* (Harris) (Hemiptera: Cicadellidae) [14,15]; Mexican bean beetle, *Epilachna varivestis* (Mulsant) (Coleoptera: Coccinellidae) [16]; and European corn borer [17]. For instance, the lowest number of *B. tabaci* eggs per leaf disc was observed on snap bean cultivar ‘Jade’, whereas cultivars ‘Gold Mine’, ‘Golden Rod’, ‘Long Tendergreen’, and ‘Royal Burgundy’ supported a significantly lower number of *B. tabaci* nymphs per leaf disc in the 2020 fall season in Georgia [13]. Pod tunneling caused by European corn borer was the most common on snap bean cultivars of ‘Gold Crop’ and ‘Pillsbury 516’, but was the least common on cultivar ‘Sungold’ [17]. The assessment of novel snap bean cultivars with multiple pest resistance traits is fundamental to providing vegetable growers with sustainable pest management strategies for snap bean production.

Little information exists on the resistance of snap bean cultivars to multiple pests in the southern United States. Therefore, one objective of our study was to assess the susceptibility of 24 local and commercially available snap bean cultivars to the infestations of multiple insect pests. Moreover, most plants rely heavily on pollinators for successful reproduction [18], and the ability of flowers to attract pollinators among different plant cultivars may vary [19,20,21,22]. Natural enemies may play an important role in suppressing populations of insect pests [23,24,25], and different cultivars of plants may have diverse impacts on natural enemies [26,27,28,29]. Therefore, we determined the population dynamics of pollinators and natural enemies on the above 24 snap bean cultivars. Additionally, because the population dynamics of insects could be affected by environmental factors [13,30,31,32], correlations between the populations of insects and climatic factors were examined.

## 2. Materials and Methods

### 2.1. Experiment Site Conditions

During 2020–2021, two field trials were implemented in an experimental field located at the Fort Valley State University New Research Farm (32°31′11″ N, 83°52′2″ W) in Fort Valley, GA, USA. In this area, the climate is humid subtropical [33] and the soil type is classified as ultisol (red clay soil) [34]. The information on the experimental site conditions was the same as that in our previous study [13].

### 2.2. Snap Bean Cultivars and Cultural Practice

The population dynamics of insect pests and beneficials (pollinators and natural enemies) were determined in 24 commercially available bush snap bean cultivars that are commonly cultivated by Georgia growers. The 24 bush snap bean cultivars included ‘Affirmed’, ‘BA0958’, ‘Barron’, ‘Bronco’, ‘Caprice’, ‘Carson’, ‘Colter’, ‘Contender’, ‘Gold Mine’, ‘Golden Rod’, ‘Greenback’, ‘Greencrop’, ‘Jade’, ‘Long Tendergreen’, ‘Maxibel’, ‘Momentum’, ‘Prevail’, ‘Provider’, ‘PV-857’, ‘Roma II’, ‘Royal Burgundy’, ‘SV1003GF’, ‘Sybaris’, and ‘Tema’ [13]. The seeds of the 24 bush snap bean cultivars were obtained from Seedway, LLC. (Hall, NY, USA) and Osborne Quality Seeds (Mount Vernon, WA, USA) [13].

Seeds of all cultivars were sown on 26 August 2020 for the fall season and on 6 April 2021 for the spring season. The seeds were sown following the local cultural practices, with a 3.8 cm sowing depth and 7.6 cm in-row spacing. The row was 3.0 m in length and the row spacing was 0.9 m. Forty-one plants were sown per row [13].

In both seasons, S-metolachlor herbicide (trade name: Dual Magnum; Syngenta Crop Protection, Inc., Greensboro, NC, USA) was applied to the experimental field at a rate of 1167.90 mL/ha the next day after sowing the seeds. No insecticide was used in the experimental field during the two seasons. Fertilizers were not applied in the experimental field during the 2020 fall season. For the 2021 spring season, the field was fertilized with N–P–K (nitrogen–phosphorus–potassium) (19–19–19) at 224 kg/ha using a farming fertilizer spreader before sowing was initiated. The experimental field was only irrigated by rainfall during the 2020 fall season. Beyond rainfall, the only irrigation that was provided to the plots was overhead irrigation once using a farm irrigation sprinkler system on the fourth week of sampling in the 2021 spring season. The cultural practices were reported in our previous study [13].

### 2.3. Experiment Design and Layout

The experiment was set up as a randomized complete block design in both seasons (2020 fall and 2021 spring). Three blocks were included in each season. Each block was comprised of two big columns, with 12 experimental plots per big column and four rows per experimental plot (Figure 1). Each experimental plot had 164 snap bean plants in an area of 8.1 m^2^. In both seasons, each of the 24 snap bean cultivars was randomly assigned to one of 24 experimental plots within each block [13].

The experimental field was situated at least 9.1 m (30 feet) from the adjacent fields (Figure 1). The inter-block distance was 5.49 m (18 feet) (Figure 1). The distance between the two big columns within each block was 2.44 m (8 feet) (Figure 1). The layout map of the experimental field in fall of 2020 was included to indicate the experimental setup (Figure 1). In the 2020 fall season, there was a peach (*Prunus persica* (L.) (Rosales: Rosaceae)) orchard on the east of the experimental field, a cotton (*Gossypium hirsutum* (L.) (Malvales: Malvaceae)) field to the north, a soybean (*Glycine max* (L.) Merrill (Fabales: Fabaceae)) field to the south, and a princess tree (*Paulownia tomentosa* (Steud.) (Lamiales: Paulowniaceae)) orchard to the west (Figure 1). The experiment was conducted in the same field using a similar layout map during the 2021 spring season except that there was only a peach orchard to the east and a princess tree orchard to the west.

### 2.4. Weather Information

During the experimental periods in both seasons, all climatic data (temperature, relative humidity, and rainfall) were acquired from a Georgia weather station at Fort Valley State University (Fort Valley, GA, USA) located approximately 3.2 km from the experimental field. Data on daily climatic variables (minimum temperature (°C), average temperature (°C), maximum temperature (°C), minimum relative humidity (%), average relative humidity (%), maximum relative humidity (%), and rainfall (mm)) were averaged for each sampling week (Table 1) [13].

### 2.5. Insect Samplings

In each season, four sampling methods (leaf-turn method [35], pan traps, yellow sticky cards, and sweep nets) were employed to collected insects from the experimental field. Insect samplings using the above four methods occurred once a week for six weeks, starting at 25 DAE (days after plant emergence) and ending at 60 DAE (Table 1). Samplings were initiated in the mornings (between 8:00 am and 10:30 am) to standardize evaluation. To reduce the impact of adjacent plots on the results, the samples were taken from the middle two rows of each plot.

The leaf-turn method was used to monitor the number of *B. tabaci* adults. Five upper leaves and five lower leaves on a plant were sampled to evaluate the number of *B. tabaci* adults, then detached and taken back to the laboratory [13]. The numbers of *B. tabaci* eggs and nymphs were then checked under a dissecting microscope (Leica EZ4 W; Leica Microsystems Inc., Buffalo Grove, IL, USA) [13].

Pan traps of three colors (blue, yellow, and white) were utilized to sample pollinators (bees, moths, and wasps). Each pan trap (top diameter: 15 cm; bottom diameter: 8.8 cm; and height: 4.0 cm) was glued on the ring hoop of a metal plant support stake (92 cm high). In each experimental plot, three pan traps with three colors (blue, yellow, and white) were randomly placed in the inter-row spaces between the first and second rows, between the second and third rows, and between the third and fourth rows. A soapy water solution at a ratio of 2.5 mL:3.785 L (soap and water) was prepared weekly, and approximately 150 mL of soapy water was added to each pan trap. The pan traps were kept in the field for 24 h. After that, all the soapy water in the three pan traps from one experimental plot was collected into a labeled plastic container (diameter: 6.9 cm; height: 8.4 cm). The plastic containers with soapy water were taken back to the laboratory. All the insects (mainly pollinators) in each plastic container were transferred into a labeled 8-dram glass vial (diameter: 25 mm; height: 95 mm) within 24 h of collection from the field. Then, about 15 mL of 70% ethanol was added into each glass vial to preserve the insect samples. The insect samples in the glass vials were stored in the laboratory for later identification.

A yellow sticky card (12.7 cm × 7.6 cm) fixed on a stake (30 cm high) was placed in the center of each experimental plot for 24 h. Thereafter, all the yellow sticky cards were covered on both sides using the plastic membrane and labeled with information on the block and cultivar. The yellow sticky cards were later taken back to the laboratory and stored in an incubator (Percival PGC-9/2; Percival Scientific, Fontana, WI, USA) at 4 °C for future identification of trapped insects. 

Four sweeps were conducted using insect sweep nets (hoop: 38 cm; handle length: 91 cm) to collect insects in each experimental plot. Insects collected from one experimental plot were placed in a labeled interlocking seal plastic bag. All the bags were taken to the laboratory and stored at 4 °C in the incubator for further identification of the insect collected.

### 2.6. Data Analysis

#### 2.6.1. Numbers of Insects

The numbers of the same insects captured on the yellow sticky cards and collected by sweep nets were pooled together for data analysis. Thus, the units of the insects included pan trap, YSC (captured on one yellow sticky card), and YSCS (YSC and collected from four sweeps (S)). In each season, data were analyzed by fitting a generalized linear mixed model to the numbers of each insect per unit. The numbers of each insect per unit were modeled by a poison or negative binomial distribution. The linear predictor involved related random effects and fixed effects (sampling weeks (Weeks 1–6), treatments (24 snap bean cultivars), and two-way interactions).

Over-dispersion was evaluated using the maximum-likelihood-based fit statistic Pearson Chi-Square/DF [36]. No evidence of over-dispersion was identified. The final statistical model used for inferences was fitted using residual pseudo-likelihood. The statistical model was fitted by the PROC GLIMMIX procedure in SAS software [37]. To avoid inflations of type I errors, comparisons were conducted by Tukey or Bonferroni adjustments.

#### 2.6.2. Correlations between the Numbers of Each Insect and Climatic Factors

In each season, the Spearman correlation analysis method was used to conduct correlation analyses between the numbers of each insect and the selected climatic factors in SAS software.

## 3. Results

### 3.1. Fall Season Experiment in 2020

During Weeks 1 and 2, the snap bean plants were in a vegetative state, then they blossomed in Weeks 3 and 4, reaching their peak blooming and producing pods in Weeks 5 and 6.

#### 3.1.1. Insect Pests

The main insect pests identified in the 2020 fall season were (1) *B. tabaci*; (2) cucumber beetles, including the spotted cucumber beetle, *Diabrotica undecimpunctata howardi* (Barber), and the striped cucumber beetle, *Acalymma vittatum* (Fab) (Coleoptera: Chrysomelidae); (3) kudzu bug, *Megacopta cribraria* (Fabricius) (Hemiptera: Plataspidae); (4) *E. varivestis*; (5) thrips, including the western flower thrips, *Frankliniella occidentalis* (Pergande), and the onion thrips, *Thrips tabaci* (Lindeman) (Thysanoptera: Thripidae); (6) tarnished plant bug, *Lygus lineolaris* (Palisot de Beauvois) (Heteroptera: Miridae); and (7) *E. fabae*. The most abundant and dominant insect pest was *B. tabaci*.

The population dynamics of *B. tabaci* eggs, nymphs, and adults on different snap bean cultivars were documented in our previous study [13]. In brief, the number of adults per leaf was significantly higher in Week 1 and was not significantly different among the 24 snap bean cultivars [13]. Overall, the cultivar ‘Jade’ supported the lowest number of eggs, whereas cultivars ‘Gold Mine’, ‘Golden Rod’, ‘Long Tendergreen’, and ‘Royal Burgundy’ harbored lower numbers of *B. tabaci* nymphs [13]. The peaks of eggs and nymphs were in Week 2 and Week 4, respectively [13].

For the 2020 fall season, there were no significant interactions between sampling weeks and snap bean cultivars for the numbers of adult cucumber beetle per YSCS, *M. cribraria* per pan trap, *E. varivestis* per YSCS, thrips per YSC, and *L. lineolaris* per YSCS (Table 2). The numbers of adult cucumber beetle per YSCS, *M. cribraria* per pan trap, *E. varivestis* per YSCS, thrips per YSC, and *L. lineolaris* per YSCS, were not significantly different among snap bean cultivars (Table 2). However, there were significant differences among sampling weeks regarding the numbers of adult cucumber beetle per YSCS, *M. cribraria* per pan trap, *E. varivestis* per YSCS, thrips per YSC, and *L. lineolaris* per YSCS (Table 2). The number of adult cucumber beetle per YSCS was significantly higher in Week 2 than in Weeks 5 and 6 (Figure 2A). The number of *M. cribraria* adults per pan trap was significantly higher in Week 3, followed by Weeks 4 and 5 (Figure 2B). There was a significantly higher number of adult *E. varivestis* per YSCS in Week 1 (Figure 2C). The numbers of adult thrips per YSC in Weeks 3 and 4 were significantly higher than in other weeks (Figure 2D). The number of adult *L. lineolaris* per YSCS in Week 4 was significantly higher than in Weeks 1–3 and 6 (Figure 2E).

A significant interaction between sampling weeks and snap bean cultivars was identified regarding the number of *E. fabae* adults per YSCS in the 2020 fall season (Table 2). The number of adult *E. fabae* per YSCS was significantly different among snap bean cultivars and among sampling weeks (Table 2). Compared to other cultivars, a significantly lower number of adult *E. fabae* per YSCS was detected on cultivars ‘Barron’, ‘Contender’, ‘Gold Mine’, ‘Golden Rod’, ‘Greencrop’, ‘Jade’, ‘Momentum’, ‘PV-857’, and ‘SV1003GF’ in Week 1; cultivar ‘Momentum’ in Week 2; cultivars ‘Carson’, ‘Golden Rod’, and ‘Greencrop’ in Week 3; cultivars ‘Contender’ and ‘PV-857’ in Week 4; cultivar ‘PV-857’ in Week 5; as well as cultivars ‘Caprice’, ‘Jade’, ‘PV-857’, and ‘SV1003GF’ in Week 6 (Table 3).

Overall, for each snap bean cultivar, the number of adult *E. fabae* per YSCS was the highest in Week 3, followed by Weeks 2 and 4–6 (Table 3).

#### 3.1.2. Pollinators

The major pollinators observed in the 2020 fall season experiment were (1) bees, including honeybees, *Apis mellifera* (L.), bumble bees, *Bombus* spp. (Hymenoptera: Apidae), and other bees; (2) moths, including the ailanthus webworm moth, *Atteva punctella* (Cramer) (Lepidoptera: Yponomeutidae); and (3) wasps, including the yellowjacket wasp, *Vespula germanica* (Fabricus) and Sphecidae wasp, *Ammophila* spp. (Hymenoptera: Vespidae). There were no significant interactions between sampling weeks and snap bean cultivars regarding the number of moths per pan trap (*F* = 0.43; df = 115, 286; *p* = 1.00) or wasps per pan trap (*F* = 1.23; df = 115, 286; *p* = 0.09). No significant differences among sampling weeks were detected in the number of moths per pan trap (*F* = 0.72; df = 5, 286; *p* = 0.61) or wasps per pan trap (*F* = 0.99; df = 5, 286; *p* = 0.42). The mean numbers of moths and wasps per pan trap over the six sampling weeks ranged from 0.023 to 0.45 and 0.45 to 1.72, respectively. There were no significant differences among snap bean cultivars regarding the number of moths per pan trap (*F* = 0.32; df = 23, 286; *p* = 1.00) or wasps per pan trap (*F* = 1.17; df = 23, 286; *p* = 0.27). The mean numbers of moths and wasps per pan trap on the 24 snap bean cultivars ranged from 0.023 to 0.63 and 0.34 to 1.65, respectively.

The interaction between sampling weeks and snap bean cultivars was not significant for the number of bees per pan trap (*F* = 0.71; df = 115, 286; *p* = 0.98). The number of bees per pan trap was not significantly different among snap bean cultivars (*F* = 1.15; df = 23, 286; *p* = 0.29). However, there were significant differences among sampling weeks regarding the number of bees per pan trap (*F* = 6.34; df = 5, 286; *p* < 0.0001). The number of bees per pan trap was significantly higher in Week 6 than in other weeks (Figure 3).

#### 3.1.3. Natural Enemies

The primary natural enemies found in snap bean plots in the 2020 fall season experiment were (1) predators, including adult *Delphastus* spp. and predator ladybird beetles (e.g., *Coccinella septempunctata* (L.) and *Harmonia axyridis* (Pallas) (Coleoptera: Coccinellidae)); (2) parasitoid wasps, including adult *Encarsia* spp. and *Eretmocerus* spp. (Hymenoptera: Aphelinidae); and (3) adult minute pirate bugs, *Orius* spp. (Hemiptera: Anthocoridae). There were no significant interactions between sampling weeks and snap bean cultivars regarding the numbers of all the discovered natural enemies above (*p* > 0.05). No significant differences were detected among sampling weeks or among snap bean cultivars in the numbers of all the discovered natural enemies above (*p* > 0.05). The mean numbers of adult *Delphastus*, predator ladybird beetles, parasitoid wasps, and *Orius* per YSCS over the six sampling weeks ranged from 0.024 to 0.39, 0.00 to 0.13, 0.00 to 0.36, and 0.00 to 0.42, respectively. The mean numbers of adult *Delphastus*, predator ladybird beetle, parasitoid wasps, and *Orius* per YSCS on the 24 snap bean cultivars ranged from 0.049 to 0.32, 0.00 to 0.20, 0.00 to 0.28, and 0.00 to 0.46, respectively.

#### 3.1.4. Correlations between the Numbers of Insects and Climatic Factors

For the 2020 fall season, negative correlations were detected for the number of adult *E. varivestis* × minimum relative humidity (r = −0.89, *p* = 0.02), the number of adult predator ladybird beetles × minimum relative humidity (r = −0.81, *p* = 0.05), the number of adult predator ladybird beetles × maximum relative humidity (r = −0.81, *p* = 0.05), and the number of adult predator ladybird beetles × rain (r = −0.93, *p* = 0.0077). Correlations were non-significant (*p* > 0.05) between the climatic factors and the numbers of other insects.

### 3.2. Spring Season Experiment in 2021

The snap bean crops had a similar crop phenology as that for the 2020 fall season. In the initial two weeks (Weeks 1 and 2), the snap bean plants underwent a vegetative phase. Afterward, they blossomed in the subsequent two weeks (Weeks 3 and 4), attaining their maximum flowering and yielding pods in Weeks 5 and 6.

#### 3.2.1. Insect Pests

In the 2021 spring season, the principal insect pests included *B. tabaci*, cucumber beetle, *E. varivestis*, *L. lineolaris*, and *E. fabae*. The populations of *E. fabae* adults were the most abundant in the field.

The population dynamic of *B. tabaci* eggs, nymphs, and adults on different snap bean cultivars was reported in an earlier study [13]. Briefly, no significant interactions were found between sampling weeks and snap bean cultivars regarding the numbers of adults, eggs, or nymphs per leaf [13]. There were no significant differences among sampling weeks or among snap bean cultivars in the numbers of *B. tabaci* adults, eggs, or nymphs per leaf [13].

No significant interactions between sampling weeks and snap bean cultivars were observed for the numbers of adult cucumber beetle, *E. varivestis*, *L. lineolaris*, and *E. fabae* per YSCS in the 2021 spring season (Table 4). The numbers of adult cucumber beetle and *E. varivestis* per YSCS were not significantly different among snap bean cultivars (Table 4). There were significant differences regarding the numbers of adult cucumber beetle and *E. varivestis* per YSCS among sampling weeks (Table 4). There were significant differences regarding the numbers of adult *L. lineolaris* and *E. fabae* per YSCS among sampling weeks and among snap bean cultivars (Table 4). The number of cucumber beetle adults per YSCS was significantly higher in Week 3 than in other weeks (Figure 4A). The number of *E. varivestis* adults per YSCS was significantly higher in Week 1, followed by Week 3 (Figure 4B). The number of *L. lineolaris* adults per YSCS was significantly higher in Weeks 3 and 4 (Figure 4C). The number of *L. lineolaris* adults per YSCS was significantly lower on cultivars ‘BA0958’, ‘Barron’, ‘Bronco’, ‘Caprice’, ‘Colter’, ‘Gold Mine’, ‘Golden Rod’, ‘Greenback’, ‘Greencrop’, ‘Long Tendergreen’, ‘Maxibel’, ‘Prevail’, ‘PV-857’, and ‘Roma II’ (Figure 5A). The number of *E. fabae* adults per YSCS was significantly higher in Week 3 (Figure 4D). The number of *E. fabae* adults per YSCS was significantly lower on cultivars ‘Barron’, ‘Gold Mine’, ‘Greenback’, ‘Greencrop’, ‘Maxibel’, ‘Prevail’, ‘PV-857’, and ‘Sybaris’ (Figure 5B).

#### 3.2.2. Pollinators

The main pollinators identified in the 2021 spring season experiment were bees (honeybees and bumble bees), moths (the ailanthus webworm moth), and wasps (the yellowjacket wasp and Sphecidae wasp). There were no significant interactions between sampling weeks and snap bean cultivars regarding the numbers of bees per pan trap (*F* = 0.83; df = 115, 286; *p* = 0.87), moths per pan trap (*F* = 0.36; df = 78, 286; *p* = 1.00), or wasps per pan trap (*F* = 0.61; df = 115, 286; *p* = 1.00). There were no significant differences among snap bean cultivars in the numbers of bees per pan trap (*F* = 0.43; df = 23, 286; *p* = 0.99), moths per pan trap (*F* = 0.02; df = 23, 286; *p* = 1.00), or wasps per pan trap (*F* = 0.66; df = 23, 286; *p* = 0.88). No significant differences among sampling weeks were detected in the number of moths per pan trap (*F* = 0.00; df = 5, 286; *p* = 1.00). However, there were significant differences among sampling weeks regarding the numbers of bees per pan trap (*F* = 5.90; df = 5, 286; *p* < 0.0001) and wasps per pan trap (*F* = 7.25; df = 5, 286; *p* < 0.0001). The number of bees per pan trap was significantly higher in Week 5, followed by Week 6 (Figure 6A). The number of wasps per pan trap was significantly higher in Weeks 2 and 6 (Figure 6B).

#### 3.2.3. Natural Enemies

The critical natural enemies discovered in the 2021 spring season experiment included adult predator ladybird beetles and *Orius* spp. No significant interactions between sampling weeks and snap bean cultivars were identified regarding the numbers of adult predator ladybird beetles and *Orius* per YSCS (*p* > 0.05). There were no significant differences among sampling weeks or among snap bean cultivars (*p* > 0.05) in the numbers of adult predator ladybird beetles and *Orius* per YSCS. The mean numbers of adult predator ladybird beetles and *Orius* per YSCS over the six sampling weeks ranged from 0.00 to 0.15 and 0.15 to 0.81, respectively. The mean numbers of adult predator ladybird beetles and *Orius* per YSCS on the 24 snap bean cultivars ranged from 0.00 to 0.17 and 0.084 to 0.72, respectively.

#### 3.2.4. Correlations between the Numbers of Insects and Climatic Factors

In the 2021 spring season, positive correlations were identified for *B. tabaci* egg infestation × minimum temperature, *B. tabaci* nymph infestation × minimum temperature, *B. tabaci* nymph infestation × maximum temperature, *B. tabaci* nymph infestation × minimum relative humidity, and *B. tabaci* nymph infestation × average relative humidity [13]. Positive correlations were also detected for the number of bees × minimum temperature (r = 1.00, *p* < 0.0001), the number of bees × average temperature (r = 0.89, *p* = 0.02), and the number of bees × maximum temperature (r = 0.94, *p* = 0.0048). Negative correlations were detected for the number of adult predator ladybird beetles × minimum temperature (r = −0.85, *p* = 0.034), the number of adult predator ladybird beetles × average temperature (r = −0.85, *p* = 0.034), and the number of adult predator ladybird beetles × maximum temperature (r = −0.85, *p* = 0.034). Correlations were non-significant (*p* > 0.05) between the climatic factors and the numbers of other insects.

## 4. Discussion

This is the first study to determine the population dynamics of multiple pests, pollinators, and natural enemies on 24 local and commercially available bush snap bean cultivars in the southern United States. Previous studies concentrated on determining the susceptibility of different snap bean cultivars to only one major pest, such as *B. tabaci* [11,12,13,30], *E. fabae* [15,38], or *L. lineolaris* [39,40]. Therefore, this study might contribute to the knowledge of the population dynamics of insect pests and beneficials as impacted by the cultivation of snap bean cultivars. The information derived from this study will help elucidate differences in the susceptibility of snap bean cultivars to different pests, and interactions among pests, pollinators, and natural enemies.

The implementation of host plant resistance has been regarded as a critical approach in pest management programs on snap bean [13,15,17]. The main pests on snap bean we identified during the two seasons included *B. tabaci*, cucumber beetle, *M. cribraria*, *E. varivestis*, thrips, *L. lineolaris*, and *E. fabae*. The susceptibility of the 24 local and commercially available snap bean cultivars to *B. tabaci* has been discussed in our previous study [13]. The results from the current study indicated that there was the lowest number of *E. fabae* adults on snap bean cultivars ‘Greencrop’ and ‘PV-857’. Previous studies demonstrated that some snap bean cultivars exhibit different levels of resistance to *E. fabae* [15,38]. It was reported that two common bean lines, ‘PI 151014’ and ‘PI 173024’, were resistant to *E. fabae* nymphal infestation [15]. The snap bean of the ‘Refugee’ type was demonstrated to be less heavily populated by *E. fabae* than cultivar ‘Green Pod’ in Ohio [38]. In this study, a significantly lower number of *L. lineolaris* adults was observed on snap bean cultivars ‘BA0958’, ‘Barron’, ‘Bronco’, ‘Caprice’, ‘Colter’, ‘Gold Mine’, ‘Golden Rod’, ‘Greenback’, ‘Greencrop’, ‘Long Tendergreen’, ‘Maxibel’, ‘Prevail’, ‘PV-857’, and ‘Roma II’ compared to other cultivars. Host plant resistance shown by some bean cultivars against *L. lineolaris* was documented in previous studies [39,40]. For instance, snap bean cultivars ‘Bountiful’ and ‘Columbia Pinto’ supported the lowest number of *L. lineolaris* adults [39]. Snap bean cultivar ‘Bountiful’ was the most tolerant to attack by *L. lineolaris* [40].

Floral resource availability during growing seasons mediates the population dynamics of pollinators [41]. In our study, the bee population in the 2020 fall season increased over the sampling weeks and reached its peak in Week 6. The populations of bees and wasps in the 2021 spring season had a similar trend and reached their peaks in Week 5 for bees and Week 6 for wasps. The population dynamics of the pollinators may be attributed to the corresponding plant phenology of the snap bean, because the snap bean plants were in vegetative growth in Weeks 1 and 2, bloomed in Weeks 3 and 4, and reached the peak of blooming and podded in Weeks 5 and 6.

Pollinators play a crucial role in the reproduction of most angiosperms [18]. Although common beans are partially autogamous, several studies have demonstrated that cross-pollination provided by insect pollinators could increase seed production by reducing abortion rates [42,43,44]. Moreover, different plant cultivars may possess various levels of attractiveness to pollinators [19,20,21,22]. For instance, two common bean varieties of ‘Carioca’ and ‘Trout’ with higher nectar and larger flowers could have more floral attractiveness to pollinators [21]. It was noted that the flower attractiveness to pollinators might vary among different soybean varieties cultivated in Brazil [22]. However, we did not find any significant differences regarding the numbers of pollinators among the 24 snap bean cultivars.

Other beneficial arthropods include natural enemies such as spiders, which are predators and play an important role in insect pest management [45,46]. Surprisingly, we only observed a few spiders using the current sampling tools (pan trap, yellow sticky cards, and sweep nets). In the future, other sampling tools may be employed to better trap spiders.

Climatic factors (e.g., temperature, relative humidity, and rainfall) are known to have a great influence on the population dynamics of pests [13,47,48,49], pollinators [50,51], and natural enemies [52,53,54]. The correlation of *B. tabaci* infestations with climatic factors has been discussed previously [13]. As for the correlations of other identified insect pests with climatic factors, we only found that the population of adult *E. varivestis* was negatively correlated with relative humidity. Other studies have directly demonstrated that the population of *E. varivestis* was greatly affected by rainfall, relative humidity, and temperature [55,56,57,58,59]. However, we are unaware of any studies assessing correlations between field populations of *E. varivestis* and climatic factors. As the dominant pest in the 2021 spring season, the population of *E. fabae* adults was not correlated with any climatic factors, which agrees with a previous study [47]. It was observed that the population of cotton leafhopper, *Amrasca biguttula biguttula* (Ishida) (Hemiptera: Cicadellidae), has non-significant correlations with all weather variables [47]. However, most previous studies reported significant correlations between leafhopper populations and climatic factors [60,61,62]. Regarding the correlations of pollinators with climatic factors, we detected that the populations of bees were positively correlated with temperature, which may be because more bees from other fields could locate the resources in our experimental field by increasing their searching ability when temperature increased. This discovery confirms the finding of a previous study indicating that the number of bees that collected nectar had a positive association with air temperature [63]. However, several previous studies detected negative correlations between the number of bees and temperature [64,65,66]. As for the correlations between the populations of natural enemies and climatic factors, the populations of predatory ladybird beetles were negatively correlated with all the selected weather variables in our experiment, which partially agrees with the findings from earlier studies [67,68]. It was reported that the populations of predatory ladybird beetles showed a positive correlation with temperature and negative correlations with relative humidity and rainfall [67].

## 5. Conclusions

In summary, cultivar ‘Jade’ harbored the lowest number of *B. tabaci* eggs, whereas cultivars ‘Gold Mine’, ‘Golden Rod’, ‘Long Tendergreen’, and ‘Royal Burgundy’ supported a significantly lower number of *B. tabaci* nymphs. Moreover, cultivars ‘Greencrop’ and ‘PV-857’ had the lowest numbers of *E. fabae* adults, while cultivars ‘BA0958’, ‘Barron’, ‘Bronco’, ‘Caprice’, ‘Colter’, ‘Gold Mine’, ‘Golden Rod’, ‘Greenback’, ‘Greencrop’, ‘Long Tendergreen’, ‘Maxibel’, ‘Prevail’, ‘PV-857’, and ‘Roma II’ harbored significantly lower numbers of *L. lineolaris* adults. Thus, the lowest numbers of both *E. fabae* and *L. lineolaris* adults were found on cultivars ‘Greencrop’ and ‘PV-857’. Cultivars ‘Gold Mine’, ‘Golden Rod’, and ‘Long Tendergreen’ had the lowest numbers of both *B. tabaci* nymphs and *L. lineolaris* adults. These snap bean cultivars could be a good option for local vegetable growers in the southern United States. The utilization of these snap bean cultivars might (1) reduce *B. tabaci*, *E. fabae*, and *L. lineolaris* populations; (2) decrease plant damage; (3) reduce insecticide use and enhance insecticide use efficiency, consequently protecting human health and the environment; (4) lower control costs; (5) minimize yield losses; and (6) increase the profitability and competitiveness of local vegetable producers in domestic and international marketplaces. In addition, the peaks for adult cucumber beetle, *M. cribraria*, and *E. fabae* were observed in Week 3, for *E. varivestis* in Week 1, for thrips in Weeks 3 and 4, and the peak for *L. lineolaris* occurred in Week 4. To obtain optimal pest management, vegetable growers could take management measures to target different pests at the corresponding peak times. The bee population reached its peak in Weeks 5 and 6, while the wasps did so in Weeks 2 and 6. Thus, vegetable growers should consider protecting pollinators when they apply pest control measures during Weeks 2, 5, and 6. Temperature, relative humidity, and rain were negatively correlated with predator ladybird beetle populations, which demonstrated that predator ladybird beetle populations will decrease, and other pest control methods need to be guaranteed when the temperature, relative humidity, and rain increase. Temperature was positively correlated with bee populations, which implied that when the temperature increases, the bee populations will increase, and vegetable growers may not need to release supplemental pollinators for pollination services.

## Figures and Tables

**Figure 1 insects-14-00230-f001:**
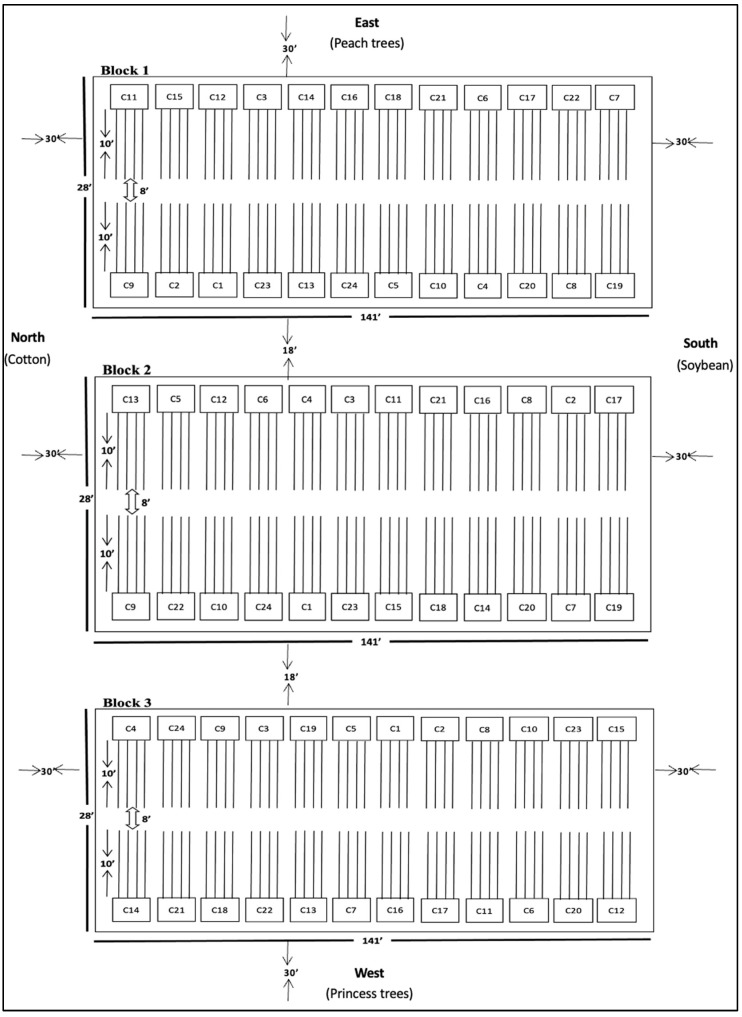
Layout of the experimental field in the 2020 fall season. Notes: When the experiment was conducted, the unit of ‘foot’ was used to measure the field, such as ‘15′’ meaning ‘15 feet’ (4.57 m). When the seeds of different snap bean cultivars were sown in the field, the number was randomly assigned to each snap bean cultivar, such as ‘C11’ meaning ‘Cultivar 11’.

**Figure 2 insects-14-00230-f002:**
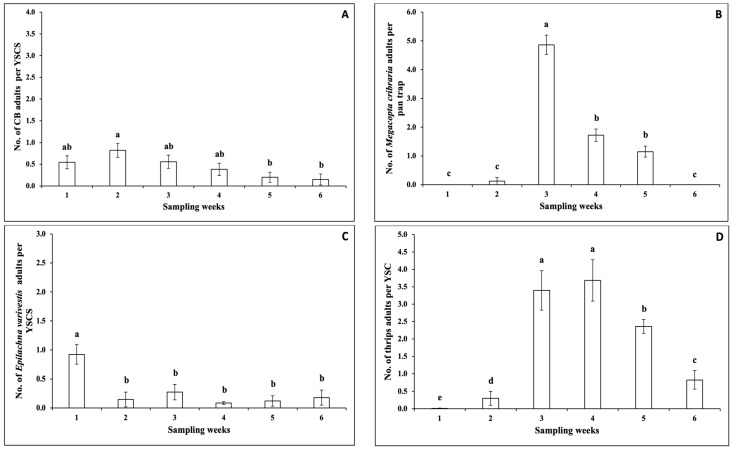
Mean (±SEM) number (No.) of adult insect pests per unit (pan trap, YSC (captured on one yellow sticky card), or YSCS (YSC and collected from four sweeps (S))) over six sampling weeks (Week 1 (25 days after plant emergence) and Weeks 2–6) in the 2020 fall season. The insect pests included (**A**) cucumber beetle (CB); (**B**) *Megacopta cribraria*; (**C**) *Epilachna varivestis*; (**D**) thrips; and (**E**) *Lygus lineolaris*. In each figure, different letters above the error bars indicate significant differences (*p* < 0.05) among sampling weeks.

**Figure 3 insects-14-00230-f003:**
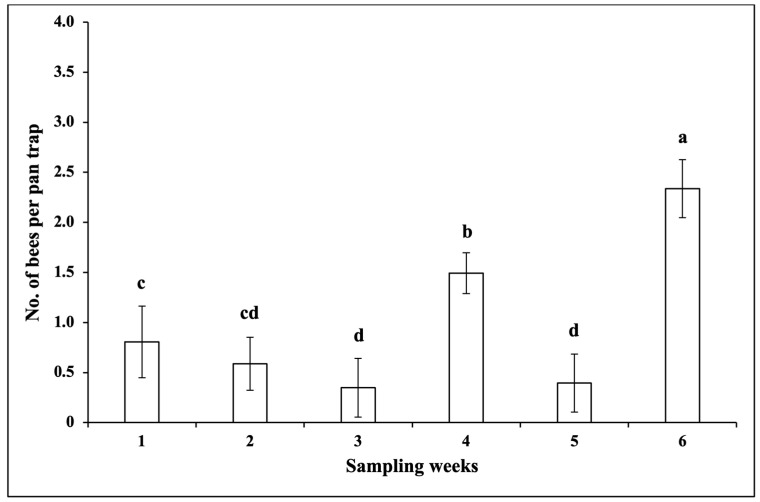
Mean (±SEM) number (No.) of bees per pan trap over six sampling weeks (Week 1 (25 days after plant emergence) and Weeks 2–6) in the 2020 fall season. Different letters above the error bars indicate significant differences (*p* < 0.05) among sampling weeks.

**Figure 4 insects-14-00230-f004:**
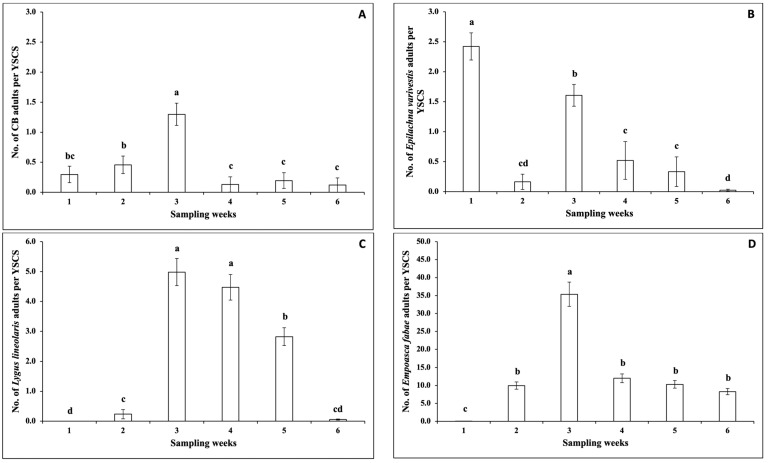
Mean (± SEM) number (No.) of adult insect pests per YSCS (captured on one yellow sticky card (YSC) and collected from four sweeps (S)) over six sampling weeks (Week 1 (25 days after plant emergence) and Weeks 2–6) in the 2021 spring season. The insect pests included (**A**) cucumber beetle (CB); (**B**) *Epilachna varivestis*; (**C**) *Lygus lineolaris*; and (**D**) *Empoasca fabae*. In each figure, different letters above the error bars indicate significant differences (*p* < 0.05) among sampling weeks.

**Figure 5 insects-14-00230-f005:**
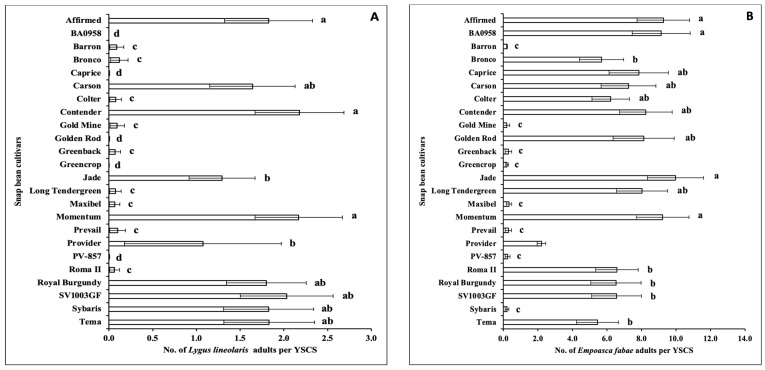
Mean (±SEM) number (No.) of adult insect pests per YSCS (captured on one yellow sticky card (YSC) and collected from four sweeps (S)) on the 24 snap bean cultivars (‘Affirmed’, ‘BA0958’, ‘Barron’, ‘Bronco’, ‘Caprice’, ‘Carson’, ‘Colter’, ‘Contender’, ‘Gold Mine’, ‘Golden Rod’, ‘Greenback’, ‘Greencrop’, ‘Jade’, ‘Long Tendergreen’, ‘Maxibel’, ‘Momentum’, ‘Prevail’, ‘Provider’, ‘PV-857’, ‘Roma II’, ‘Royal Burgundy’, ‘SV1003GF’, ‘Sybaris’, and ‘Tema’) in the 2021 spring season. The insect pests included (**A**) *Lygus lineolaris* and (**B**) *Empoasca fabae*. In each figure, different letters beside the error bars indicate significant differences (*p* < 0.05) among snap bean cultivars.

**Figure 6 insects-14-00230-f006:**
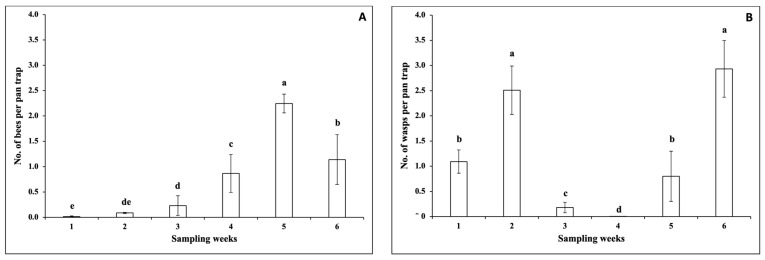
Mean (±SEM) number (No.) of bees (**A**) and wasps (**B**) per pan trap over six sampling weeks (Week 1 (25 days after plant emergence) and Weeks 2–6) in the 2021 spring season. In each figure, different letters above the error bars indicate significant differences (*p* < 0.05) among sampling weeks.

**Table 1 insects-14-00230-t001:** Mean (±SEM) temperature (°C), relative humidity (%), and rainfall (mm) for each of the six sampling weeks (Week 1 (25 days after plant emergence) and Weeks 2–6) in the 2020 fall and 2021 spring seasons in Fort Valley, GA, USA.

Sampling Weeks (Dates)	Temperature (°C)			Relative Humidity (%)			Rainfall (mm)
	Minimum	Average	Maximum	Minimum	Average	Maximum	
2020 fall							
1 (28/Sep.–04/Oct.)	10.98 (±0.65)	17.42 (±0.66)	23.87 (±1.34)	41.38 (±6.48)	73.76 (±5.37)	94.88 (±3.37)	0.11 (±0.00)
2 (05/Oct.–11/Oct.)	18.04 (±1.88)	23.06 (±1.08)	28.07 (±1.50)	55.30 (±4.67)	83.65 (±2.27)	99.27 (±0.56)	2.83 (±0.08)
3 (12/Oct.–18/Oct.)	12.87 (±2.19)	19.44 (±2.14)	26.02 (±2.18)	48.12 (±1.74)	77.35 (±2.93)	96.39 (±1.93)	0.04 (±0.00)
4 (19/Oct.–25/Oct.)	16.95 (±0.81)	21.70 (±0.45)	26.44 (±0.66)	58.61 (±2.54)	86.48 (±2.56)	98.77 (±0.77)	1.92 (±0.06)
5 (26/Oct.–01/Nov.)	13.54 (±8.61)	19.01 (±9.84)	24.49 (±11.14)	51.79 (±5.72)	80.07 (±3.95)	96.51 (±2.02)	0.51 (±0.01)
6 (02/Nov.–08/Nov.)	9.21 (±4.83)	15.83 (±3.36)	22.45 (±2.15)	42.08 (±7.71)	70.13 (±6.39)	95.21 (±1.89)	0.83 (±0.03)
2021 spring							
1 (10/May–16/May)	11.71 (±2.88)	18.19 (±1.13)	24.57 (±1.91)	45.02 (±8.31)	74.01 (±5.12)	98.61 (±0.86)	5.33 (±0.17)
2 (17/May–23/May)	14.16 (±1.01)	21.66 (±0.29)	28.43 (±1.05)	37.52 (±2.50)	66.65 (±1.22)	97.20 (±1.27)	0.15 (±0.01)
3 (24/May –30/May)	16.49 (±1.83)	23.78 (±0.89)	30.87 (±1.94)	33.80 (±2.10)	66.67 (±2.03)	96.76 (±1.01)	2.47 (±0.09)
4 (31/May –06/June)	16.97 (±2.09)	23.43 (±0.73)	30.30 (±1.25)	40.43 (±2.04)	72.20 (±2.49)	99.64 (±0.20)	0.62 (±0.02)
5 (07/June–13/June)	20.30 (±0.60)	24.29 (±0.34)	31.19 (±0.78)	52.10 (±3.42)	85.38 (±2.36)	99.71 (±0.19)	4.79 (±0.12)
6 (14/June–20/June)	19.41 (±1.09)	24.91 (±0.64)	30.97 (±2.60)	47.06 (±7.37)	74.54 (±5.51)	96.50 (±1.50)	9.40 (±0.28)

**Table 2 insects-14-00230-t002:** Type III tests of fixed effects (sampling week (Week) × snap bean cultivar (Cultivar), Week, and Cultivar) for the numbers of adult cucumber beetle per YSCS (captured on one yellow sticky card (YSC) and collected from four sweeps (S)), *Megacopta cribraria* per pan trap, *Epilachna varivestis* per YSCS, thrips per YSC, *Lygus lineolaris* per YSCS, and *Empoasca fabae* per YSCS in the 2020 fall season.

Insect Pests	Fixed Effects	*F*	df	*p*
Cucumber beetle				
	Week × Cultivar	0.29	115, 286	1.00
	Week	4.39	5, 286	0.0007
	Cultivar	0.26	23, 286	1.00
*Megacopta cribraria*				
	Week × Cultivar	0.73	115, 286	0.97
	Week	88.61	5, 286	<0.0001
	Cultivar	0.45	23, 286	0.99
*Epilachna varivestis*				
	Week × Cultivar	0.21	115, 276	1.00
	Week	5.72	5, 276	<0.0001
	Cultivar	0.37	23, 276	1.00
Thrips				
	Week × Cultivar	1.09	115, 286	0.06
	Week	70.18	5, 286	<0.0001
	Cultivar	0.27	23, 286	1.00
*Lygus lineolaris*				
	Week × Cultivar	0.62	115, 286	1.00
	Week	28.79	5, 286	<0.0001
	Cultivar	0.37	23, 286	1.00
*Empoasca fabae*				
	Week × Cultivar	1.75	115, 286	<0.0001
	Week	14.19	5, 286	<0.0001
	Cultivar	2.10	23, 286	0.0028

**Table 3 insects-14-00230-t003:** Mean (±SEM) number of *Empoasca fabae* adults per YSCS (captured on one yellow sticky card (YSC) and collected from four sweeps (S)) on 24 snap bean cultivars (‘Affirmed’, ‘BA0958’, ‘Barron’, ‘Bronco’, ‘Caprice’, ‘Carson’, ‘Colter’, ‘Contender’, ‘Gold Mine’, ‘Golden Rod’, ‘Greenback’, ‘Greencrop’, ‘Jade’, ‘Long Tendergreen’, ‘Maxibel’, ‘Momentum’, ‘Prevail’, ‘Provider’, ‘PV-857’, ‘Roma II’, ‘Royal Burgundy’, ‘SV1003GF’, ‘Sybaris’, and ‘Tema’) over six sampling weeks (Week 1 (25 days after plant emergence) and Weeks 2–6) in the 2020 fall season. Means followed by different lowercase letters in each column are significantly different (*p* < 0.05) among snap bean cultivars within each sampling week. Means followed by different uppercase letters in each row are significantly different (*p* < 0.05) among sampling weeks within each snap bean cultivar.

Cultivars	Sampling Weeks						Average Counts among Weeks
	1	2	3	4	5	6	
Affirmed	0.99 ± 0.58 bD	6.63 ± 1.53 bcC	16.24 ± 2.51 aA	8.28 ± 1.73 abB	8.25 ± 1.78 aB	8.29 ± 1.73 cdB	6.25 ± 0.90
BA0958	0.66 ± 0.47 bcD	9.28 ± 1.84 abA	4.97 ± 1.32 fgC	4.64 ± 1.27 cC	5.61 ± 1.43 cB	6.63 ± 1.53 dB	4.16 ± 0.70
Barron	0.33 ± 0.33 cD	5.30 ± 1.36 cdB	11.60 ± 2.08 bA	4.64 ± 1.27 cB	4.29 ± 1.24 dBC	3.98 ± 1.17 fgC	3.41 ± 0.71
Bronco	2.32 ± 0.89 aD	4.31 ± 1.22 deC	8.29 ± 1.73 dA	8.28 ± 1.73 abA	7.59 ± 1.69 bA	6.96 ± 1.57 dB	5.73 ± 0.75
Caprice	1.33 ± 0.67 bD	10.93 ± 2.01 aA	6.96 ± 1.57 eB	6.30 ± 1.49 bcB	4.79 ± 1.31 cdC	3.32 ± 1.07 gC	4.63 ± 0.68
Carson	1.33 ± 0.67 bB	4.97 ± 1.32 cdA	4.31 ± 1.22 gA	4.97 ± 1.32 cA	4.95 ± 1.34 cdA	4.97 ± 1.32 efA	3.88 ± 0.59
Colter	0.99 ± 0.58 bD	5.63 ± 1.41 cC	8.29 ± 1.73 dA	6.96 ± 1.57 bB	6.44 ± 1.55 bcB	5.97 ± 1.45 deC	4.80 ± 0.72
Contender	0.33 ± 0.33 cD	4.64 ± 1.27 cdB	5.64 ± 1.41 fA	3.31 ± 1.07 dC	3.96 ± 1.19 dBC	4.64 ± 1.27 fB	2.83 ± 0.60
Gold Mine	0.33 ± 0.33 cE	3.31 ± 1.07 eD	12.60 ± 2.17 abA	9.28 ± 1.84 aB	7.26 ± 1.65 bC	5.31 ± 1.36 efC	4.11 ± 0.84
Golden Rod	0.00 ± 0.00 cD	9.28 ± 1.84 abA	2.32 ± 0.89 gC	8.28 ± 1.73 abA	6.44 ± 1.55 bcB	4.64 ± 1.27 fB	1.61 ± 1.06
Greenback	0.66 ± 0.47 bcD	7.29 ± 1.61 bcB	9.94 ± 1.91 cA	4.97 ± 1.32 cC	4.95 ± 1.34 cdC	4.97 ± 1.32 efC	4.23 ± 0.71
Greencrop	0.33 ± 0.33 cC	4.64 ± 1.27 cdA	3.98 ± 1.17 gB	4.97 ± 1.32 cA	4.79 ± 1.31 cdA	4.64 ± 1.27 fA	2.95 ± 0.62
Jade	0.33 ± 0.33 cD	5.30 ± 1.36 cdB	9.28 ± 1.84 cdA	6.96 ± 1.57 bB	4.95 ± 1.34 cdB	2.98 ± 1.01 gC	3.44 ± 0.71
Long Tendergreen	0.66 ± 0.47 bcD	7.29 ± 1.61 bcBC	8.29 ± 1.73 dB	6.96 ± 1.57 bC	8.91 ± 1.86 aB	10.94 ± 2.01 bA	5.46 ± 0.88
Maxibel	0.99 ± 0.58 bD	4.97 ± 1.32 cdC	7.29 ± 1.61 eB	6.96 ± 1.57 bB	7.92 ± 1.74 abB	8.95 ± 1.80 cA	5.09 ± 0.76
Momentum	0.33 ± 0.33 cD	2.32 ± 0.89 fC	13.26 ± 2.24 abA	5.30 ± 1.36 cB	5.78 ± 1.46 cB	6.30 ± 1.49 deB	3.53 ± 0.74
Prevail	0.66 ± 0.47 bcC	4.64 ± 1.27 cdAB	8.29 ± 1.73 dA	6.63 ± 1.53 bA	7.26 ± 1.65 bA	7.96 ± 1.69 cdA	4.61 ± 0.76
Provider	1.66 ± 0.75 bC	7.29 ± 1.61 bcA	7.63 ± 1.65 deA	6.96 ± 1.57 bA	6.27 ± 1.52 bcA	5.64 ± 1.41 eB	5.30 ± 0.73
PV-857	0.00 ± 0.00 cD	6.30 ± 1.49 cB	8.29 ± 1.73 dA	3.65 ± 1.12 dC	3.47 ± 1.10 eC	3.32 ± 1.07 gC	1.39 ± 0.59
Roma II	0.66 ± 0.47 bcD	3.64 ± 1.12 deC	7.95 ± 1.69 deB	4.64 ± 1.27 cC	8.75 ± 1.84 aB	12.93 ± 2.21 aA	4.63 ± 0.77
Royal Burgundy	1.33 ± 0.67 bC	4.31 ± 1.22 deB	12.26 ± 2.14 bA	5.63 ± 1.41 bcB	4.79 ± 1.31 cdB	3.98 ± 1.17 dB	4.41 ± 0.65
SV1003GF	0.33 ± 0.33 cD	6.63 ± 1.53 bcB	11.27 ± 2.04 bcA	3.98 ± 1.17 cdC	3.63 ± 1.13 dC	3.32 ± 1.07 gC	3.24 ± 0.68
Sybaris	0.99 ± 0.58 bC	7.95 ± 1.69 bB	14.58 ± 2.36 aA	6.96 ± 1.57 bB	7.92 ± 1.74 abB	8.95 ± 1.80 cB	6.18 ± 0.89
Tema	2.98 ± 1.01 aD	6.30 ± 1.49 cB	11.27 ± 2.04 bcA	4.97 ± 1.32 cC	5.12 ± 1.36 cC	5.30 ± 1.36 efC	5.51 ± 0.72
**Average counts among cultivars**	0.47 ± 0.29	5.61 ± 0.53	8.18 ± 0.74	5.79 ± 0.54	5.79 ± 0.54	5.57 ± 0.53	

**Table 4 insects-14-00230-t004:** Type III tests of fixed effects (sampling week (Week) × snap bean cultivar (Cultivar), Week, and Cultivar) for the numbers of adult insect pests (cucumber beetle, *Epilachna varivestis*, *Lygus lineolaris*, and *Empoasca fabae*) per YSCS (captured on one yellow sticky card (YSC) and collected from four sweeps (S)) in the 2021 spring season.

Insect Pests	Fixed Effects	*F*	df	*p*
Cucumber beetle				
	Week × Cultivar	0.34	115, 284	1.00
	Week	10.38	5, 284	<0.0001
	Cultivar	0.25	23, 284	1.00
*Epilachna varivestis*				
	Week × Cultivar	0.43	115, 284	1.00
	Week	7.06	5, 284	<0.0001
	Cultivar	0.38	23, 284	1.00
*Lygus lineolaris*				
	Week × Cultivar	0.77	101, 284	0.94
	Week	3.72	5, 284	0.00
	Cultivar	4.64	23, 284	<0.0001
*Empoasca fabae*				
	Week × Cultivar	0.91	107, 284	0.71
	Week	82.99	5, 284	<0.0001
	Cultivar	11.00	23, 284	<0.0001

## Data Availability

The data presented in this study are available on request from the corresponding author.

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
