# Peer review of "Population Dynamics of Insect Pests and Beneficials on Different Snap Bean Cultivars"

_insects, 2023, doi:10.3390/insects14030230_

Round 1
Reviewer 1 Report
In the current manuscript, Li et al. assessed the distribution of common bean pests and their natural enemies during two field growing seasons in Georgia. The authors have done excellent work in conducting various studies and their presentations. However, to improve the paper’s readability, I recommend removing Tables 1, 2, and 4 and including them as supplementary files.
Author Response
Responses to comments from Reviewer #1:
In the current manuscript, Li et al. assessed the distribution of common bean pests and their natural enemies during two field growing seasons in Georgia. The authors have done excellent work in conducting various studies and their presentations. However, to improve the paper’s readability, I recommend removing Tables 1, 2, and 4 and including them as supplementary files.
Response: Thank you for your suggestions. After we discussed it, we would like to keep them in the paper.
The rationale for keeping Table 1 is that we provided information on Temperature, Relative humidity, and rainfall in Table 1. We conducted correlations between populations of insects and the above three weather variables. The readers would be interested in knowing the range of each variable in the experiment. What is more, we provided information on sampling weeks (dates). That information is important for readers to know when we conducted the samplings.
The reason why we would like to keep Tables 2 and 4 is that those two tables listed the Type III tests of fixed effects [sampling week (Week) × snap bean cultivar (Cultivar), Week, and Cultivar] for the numbers of insects, including F, df, and P. That information is important for readers to know how the effects of those fixed effects on the number of insects were significant. For the non-significant effects of natural enemies, we did not provide that information.
Reviewer 2 Report
Please see attached review report.

Author Response
Responses to comments from Reviewer #2:
The manuscript titled, “Population dynamics of insect pests and beneficials on different snap bean cultivars”, by Li et al is a well written and informative study. The first paragraph of introduction provides a good overview of the snap bean economic impact. Overall, the experimental design is appropriate, and results are explained well. However, there are some missing information and sources of confusion, that needs to be addressed.
These are as follows:
Abstract: Did the ladybeetles did not show any population peaks?
Response: Yes. That is correct. In both seasons, no significant differences were detected among sampling weeks on the numbers of predator ladybird beetle. So, the ladybeetles did not show any population peaks. That information was mentioned on Lines 298-300 for 2020 fall and Lines 389-391 for 2021 spring.
Ln 72: replace and cultivars ‘Gold Mine’…. with whereas cultivars ‘Gold Mine’.
Response: We have revised it.
Ln 71 and 74: leaf disk versus disc
Response: We have revised it.
Ln 82: Is yield affected due to pollinators? Why are pollinators important in this system? This needs to be explained.
Response: First, based on the results, there was no significant difference among snap bean cultivars on the number of the pollinators discovered in this study. It means that all the snap bean cultivars attracted similar number of pollinators. So, we do not know whether the pollinators affect the yield. Second, the objective of this study was to determine whether the ability of flowers to attract pollinators among different snap bean cultivars varies. We may evaluate the effect of pollinators on yield in future.
We have added one statement about the importance of the pollinators in the system on Lines 82-83.
Ln 84: similarly, why are natural enemies important for this system? This hasn’t been explained.
Response: We have revised it.
Ln 95: what are ‘energy trees’? Is there no scientific name associated with this common name?
Response: We have revised it.
Ln 111-112: What was the length of the buffer between plots?
Response: there was not buffer between plots. The reason why we did not include buffer between plots was that there were two big columns in one block, 12 cultivars in one big column, and 4 plots of each cultivar. It means that there were 40 plots in one big column. If we also included buffer plots, it would make the big column longer in one block. To better control variables in one block, we try to make each block smaller. Even we did not include a buffer between plots, insect collection only occurred in the two center rows of each plot (cultivar) to avoid the inferences from neighboring plots. We have added one statement in the first paragraph of section of 2.5.
Ln 114: Dual Magnum should be in paranthesis and herbicide S-metolachlor should be outside as trade name can vary but active ingredient provides a more general information.
Response: We have revised it.
Ln 126: Where the three blocks the replicates? If so, needs to be specified clearly. It is a suggestion to add a plot map figure to clarify the experimental design.
Response: the experiment design was randomized complete block design. There were three blocks. In general, when we analyze data, someone regards “block” as “replicate”. However, they are different. We have added one plot map to clarify the experimental design in Figure 1.
Ln 147: ‘t’ is missing from plant.
Response: We have revised it.
Ln 155: white pan traps were placed between the four rows? Not clear.
Response: We have revised it.
Ln 156: I am unclear about the frequency/distance of pan traps from each other.
Response: There were three pan traps in each plot. One is blue, one is yellow, and one is white. There were four rows in each plot and three inter-row spaces. So, each of the three pan traps was randomly placed in one of the inter-row spaces. The distance of pan traps from each other was about row width (0.9 m). We have added one statement to make it clear.
Ln 159: Was the soapy water in three pans pooled? If so, what was the rationale of having 3 separate pan colors?
Response: Yes. The soapy water in three pans from each plot was pooled. The rationale of having 3 separate pan colors is to attract all pollinators through the three most common colors used for trapping pollinators.
Table 1: Check formatting as the word Rainfall doesn’t fit in the column correctly. There is a line going through it.
Response: We have revised it.
Ln 167: I am unclear about placement distance/frequency of sticky cards also.
Response: A yellow sticky card fixed on a stake was placed in the center of each experimental plot. The row width was 0.9 m. So, the distance of two sticky cards in neighboring plots was 3.6 m.
Ln 194: Why hasn’t a comparison with crop phenology and insect number been included in addition to climatic factors, since in discussion, floral resources have been mentioned to explain the trends in pollinator populations.
Response: We have discussed the population dynamics of pests and beneficials among different weeks in the paper. Different weeks represent crop phenology. To make it clear, we have added the statement of crop phenology before we discussed the results for the pests and beneficials. So, there were corresponding comparisons of crop phenology (reported as Weeks in the paper) and insect number. To make it simple, we keep using “Weeks” in the paper when we reported the results.
Ln 203: Scientific name and taxonomic authority is missing at several locations, and this is one of those examples where kudzu bug is mentioned by its common name only.
Response: We have revised it.
Ln 216 – 231: In this section, I still have issues with comparing with week 1-6 rather than mention the month or crop phenology instead. Week 1-6 provides no context.
Response: To make it clear, we have added one statement about crop phenology on each corresponding weeks (Weeks 1-6) at the beginning of results for each season before we presented the results of pests and beneficials.
Figures 1, 4, 5: The fonts need to be bigger, especially the letters showing significant difference in means.
Response: We have revised it.
Ln 268 – 271 and Ln 295-296: taxonomic authority is missing in this section for most insects. Also, when species is unknown, it should be written as spp. And not sp. Please check the entire manuscript for such errors.
Response: We have revised it.
Ln 387: Which lady beetles species were detected? Also, please correct Orius sp. to Orius spp.
Discussion: There are some scientific names and some common names. I suggest being consistent and only use scientific names after first mention.
Response: We have revised “lady beetles species” and “Orius sp.”. As for using scientific names and common names in Discussion, the common names were only used for cucumber beetles and thrips. Because there were two kinds of cucumber beetles, such as stripped cucumber beetle and spotted cucumber beetle. It is the same thing for thrips. There were two kinds of thrips, such as western flower thrips and onion thrips. Except cucumber beetle and thrips, other insects all used scientific names in Discussion.
Ln 435: These two snap bean cultivars were not included in the present study, it seems. However, is there any association by lineage?
Response: Yes. The two snap bean cultivars of ‘Bountiful’ and ‘Columbia Pinto’ were not included in the present study. ‘Columbia Pinto’ is not commercially available right now. ‘Bountiful’ was not a popular cultivar in Georgia because it was not recommended by extension specialist in University of Georgia. We may consult some breeders about whether there is any association by lineage.
Ln 450: It is great that the floral resource factor has been acknowledged here when explaining your observations regarding pollinators. However, circling back to my original concern earlier, why not include crop phenology information rather than Week 1, 2…etc., which would make the trend much more relatable?
Response: We have revised it. To make it clear, we have added one statement about crop phenology on each corresponding weeks (Weeks 1-6) at the beginning of results for each season before we presented the results of pests and beneficials. To make it simple, we still keep using “Weeks” in the paper.
I liked the conclusion section, especially the recommendation that with high rainfall, temperature, and RH, snap bean growers will be unable to count on lady beetle predation and need to rescue the crop accordingly.
Response: Thank you for your comments.
Reviewer 3 Report
Dear Authors ,
I read carefully your sumitted mns Insects-2161923, titled "Population dinamics of Insects pests and beneficials on different snap bean cultivars" and it could be considered a nice contribution to understand dynamics of populations of both pests and beneficials on crops and how they might provide new IPM strategies to grovers. However, the mns can be improved in some parts of the text , by adding more details in M&M and Results, more comments in Discussion , and a period (line461 to 486) has to be re-written clearly and avoiding repetitions. See, please, my notes in the revised attached file and follow my suggestions. They are important to raise the quality presentation of the manuscript.

Author Response
Responses to comments from Reviewer 3:
Line 108: I suggest to record briefly only the main details about the sources of the snap bean cvs utilized and to put also the reference.
Response: We have revised it.
Line 113: As recorded in the previous reviewer ‘note.
Response: We have revised it.
Line 116: How the fertilizer was distributed ? Method.
Response: We have revised it.
Line 129: The experimental design has to be recorded in details. It could not be “a copy” of a previous study, otherwise the Authors have to justfy the use of a previosus work plan already published.
Response: We have revised it.
Line 189: A reference?
Response: We have revised it.
Line 195: A reference?
Response: We have added one phrase to state that Spearman correlation analysis method was conducted in SAS software. Spearman correlation analysis method is one of methods to determine correlation between two factors, which we learned from coursework. We are not aware of any references to document how to conduct this analysis step by step.
Line 215: I don’t understand why theAA. include here some data referring to B.tabaci, that have already been published in a quite recent paper (doi.org/10.1016/j.cropro.2022.106022 ). If there are some interesting results that need to be compared with those provided in the present study (and for different groups of insects), I suggest to draft tables or graphics in order to show differences or similarities revealed between B.tabaci and other insects….and, then, provide suitable comments.
Response: The rationale for including the previous data about B. tabaci was to present information of all the insects discovered in the field. All the insects included B. tabaci even we reported it previously. So, we only included a summary of previous data from B. tabaci. In the Conclusion, we summarized some cultivars which had the least numbers of both B. tabaci nymphs and L. lineolaris adults. So, the growers can use those cultivars with fewer problems from B. tabaci nymphs and L. lineolaris adults.
Line 313: No any table or graphic to show such results?
Response: We did not include any tables or graphics to show the results of “Correlations were non-significant (P > 0.05) between the climatic factors and the numbers of other pests, pollinators, or natural enemies”. The rationale was that the correlations were not significant. So, we did not present the results in tables or graphics.
Line 323: See, please, my previous note at line 215.
Response: The rationale for including the previous data about B. tabaci was to present information of all the insects discovered in the field. All the insects included B. tabaci even we reported it previously. So, we only included a summary of data from B. tabaci. In the Conclusion, we summarized some cultivars which had the least numbers of both B. tabaci nymphs and L. lineolaris adults. So, the growers can use those cultivars with less B. tabaci nymphs and L. lineolaris adults.
Line 351: Delete this line in the caption, because the species names have been recorded in i the graphics.
Response: Thank you for your comments. In general, the figures should stand alone and need to include all the necessary information. Even we mentioned the species names in the graphics, we still would like to include the pest names in the figure caption. Especially for the first pest of cucumber beetle, the abbreviation of “CB” was used in the graphic. So, we need to list the whole name (cucumber beetle) for the abbreviation of “CB” in the figure caption. Otherwise, the audiences do not know the meaning of “CB”.
Line 357-359: Delete the names of cvs in the caption of Figure 4.
Response: Thank you for your comments. In general, the figures should stand alone and need to include all the necessary information. Even we mentioned the cultivar names in the graphics, we still would like to specify the cultivar information in the figure caption.
Line 407: Any table or graphic to show these results?
Response: We did not include any tables or graphics to show the results of “Correlations were non-significant (P > 0.05) between the climatic factors and the numbers of other pests, pollinators, or natural enemies.” The rationale was that the correlations were not significant. So, we did not present the results in tables or graphics.
Line 413: might contribute
Response: We have revised it.
Line 458-459: Could the AA. provide any interesting comment about this finding?
Response: We have revised it.
Line 486: The period from line 461 to 486 has to be re-written clearly and avoiding many repetitions. Also the AA. have to explain/comment the effect of a few interesting results on the target insects.i.e. the negative effects of temperature on bees and natural enemies.
Response: We have revised it.
Line 507: I ask the AA. to better explain this concept because one of the most important aim of IPM is to provide to growers strategies in order to avoid the “peak times” of pests which mean damages, loss of production and high implementation of insecticide use.
Response: We have revised it. In the sentence right above this sentence, it stated that “the peaks for adult cucumber beetle, kudzu bug, and E. fabae were observed on Week 3; for E. varivestis on Week 1; as well as for thrips on Weeks 3 and 4; while the peak for L. lineolaris occurred on Week 4”. We have added one statement to explain crop phenology in different weeks in Results before we discussed the results for the pests and beneficials. So, the growers would know the populations of adult cucumber beetle, kudzu bug, and E. fabae will reach the highest number on Week 3 during which the snap bean plants were blossoming. It is time for growers to take measures.
Reviewer 4 Report
Review of MS – Insect -2161923 “Population Dynamics of Insect Pests and Beneficials on Different Snap Bean Cultivars”
The MS addresses a topic concerning a comparison of the insect population dynamics of different cultivars of Snap Bean. In particular, the authors in the MS highlight both the pest insects and the beneficials present on the hosts and their dynamic. The MS so addresses a complex but very interesting subject for a review process in good detail. A positive consideration of the MS presented is linked to the considerable effort adopted by the authors who have planned a remarkable series of sampling on 24 cultivars. Understandably, this type of monitoring on the plan and with traps is greatly affected by the ability to coordinate the activities and the manual skills of the operators.
a) Several considerations arise for the MS evaluated and the first with reference to the climatic data used in the correlations, it is not clear to me how they were selected as often the values of different parameters are highly correlated and the choice of one or the other is not random. There are analyzes to reduce collinearity as principal component analysis (PCA) is one of the most common.
b) In the results it would be appropriate to indicate in a table the abundances of the species for each cultivar for each individual species investigated;
c) The bibliographic references are to be checked carefully as I have repeatedly found inconsistencies.
Specifics
Line 42. You can express the value in tons;
Lines 50-53. You can simplify this sentence as too general;
Line 50-60. Introduce a more specific bibliography on viruses involved in the snap bean. From my analysis of the bibliography reported by you, there are no percentages of reduction due to snap bean viruses;
Line 86 Correlation or relationship?;
Line95. What are energy trees?;
Lines 95-97. This sentence is superfluous. It is understandable that both trees and seasonal crops fall within the study area. I really don't understand the importance of this sentence. To simplify, you can insert a drawing with the distribution scheme of the parcels and neighboring areas;
Lines 100-103. Although this crop is autogamous sometimes visited by pollinating insects. It seems the value of pollination services to common bean yield is not known (see: 10.7717/peerj.10102).
Line 111. Specify the number of plants for each single experimental plot. It seems to be 2.7 m2, correct?;
Lines 132-133. Specify the distance between the climatic station and the study area. Also are the experimental parcels and the climatic station in the same altimetry?
Line 151. Why did you choose to use more colors in the traps? Don't you think that for every siglolo experimental plot these traps are excessive as they may have conditioned the densities of some species?;
Line 162. Are you sure they're mostly pollinators?
Line 173. Specify whether the choice of the sweeps time for each single experimental particle was causal;
Line194. Each insect or each insect species?
Line203. Are the authors with kudzu bug referring to Megacopta cribraria'?
Lines 295–296. Encarsia sp. and Eretmocerus sp.;
In paragraph 3.2.1. I don't understand why the results of a previous paper are reported;
Line422-423. Remove this sentence as it was specified in the previous lines.
Line454-456. I do not understand the need to justify through this reference as the mango. The species are very different and also in mango pollination is closely linked to pollinating insects.
Line 470-471. Nel riferimento bibliografico citato <<39>> non c’è nessun riferimento ad Empoasca fabae;
498-499. This statement "suppress B. tabaci, E. fabae, and L. lineolaris populations" seems excessive to me. A reduction in the populations of these pests is more likely.
Author Response
Responses to the comments from Reviewer 4:
Review of MS – Insect -2161923 “
Population Dynamics of Insect Pests and Beneficials on Different Snap Bean Cultivars”
The MS addresses a topic concerning a comparison of the insect population dynamics on different cultivars of Snap Bean. In particular, the authors in the MS highlighted the dynamics of both the pest insects and the beneficial insects present on the host snap bean cultivars. The MS so addresses a complex but very interesting subject for a review process in good detail. A positive consideration of the MS presented is linked to the considerable effort adopted by the authors who have planned a remarkable series of sampling on 24 cultivars. Understandably, this type of monitoring on the plan and with traps is greatly affected by the ability to coordinate the activities and the manual skills of the operators.
- a) Several considerations arise for the MS evaluated and the first with reference to the climatic data used in the correlations, it is not clear to me how they were selected as often the values of different parameters are highly correlated and the choice of one or the other is not random. There are analyzes to reduce collinearity as principal component analysis (PCA) is one of the most common.
Response: Thank you for your suggestions. Yes. It is a good method to reduce collinearity using PCA. We may consider using this method in future study. However, it is common to select climatic factors to correlate with insect populations in previous publications (see below), particularly as climatic factors affect both plant growth and insect development.
Huff, F.A. Relation between leafhopper influxes and synoptic weather conditions. J. Appl. Meteorol. Climatol. 1963, 2, 39–43. https://doi.org/10.1175/1520-0450(1963)002<0039:RBLIAS>2.0.CO;2.
Showers, W.B.; De Rozari, M.B.; Reed, G.L.; Shaw, R.H. Temperature-related climatic effects on survivorship of the European corn borer. Environ. Entomol. 1978, 7, 717–723.
da Silva, A.G., Boiça Júnior, A.L., da Silva Farias, P.R., de Souza, B.H.S., Rodrigues, N.E. L., Carbonell, S.A.M., 2019. Common bean resistance expression to whitefly in winter and rainy seasons in Brazil. Sci. Agric. 76, 389–397. https://doi:10.1590/1678-992x-2017-0434.
Keshan, M.A., Khan, M.A., Ali, S., Arshad, M., 2015. Correlation of conducive environmental conditions for the development of whitefly, Bemisia tabaci population in different tomato genotypes. Pakistan J. Zool. 47, 1511–1515.
Pathania, M., Verma, A., Singh, M., Arora, P.K., Kaur, N., 2020. Influence of abiotic factors on the infestation dynamics of whitefly, Bemisia tabaci (Gennadius 1889) in cotton and its management strategies in North-Western India. Int. J. Trop. Insect Sci.40, 969–981. https://doi.org/10.1007/s42690-020-00155-2.
- b) In the results it would be appropriate to indicate in a table the abundances of the species for each cultivar for each individual species investigated;
Response: Thank you for your suggestions. The objective of this study is providing some promising pest-resistant snap bean cultivars for growers. We have already summarized the promising cultivars against pests in Conclusion. So, having a table to list abundances of the species for each cultivar for each individual species may not support this objective.
- c) The bibliographic references are to be checked carefully as I have repeatedly found inconsistencies.
Response: We have revised it.
Specifics
Line 42. You can express the value in tons;
Response: We have revised it.
Lines 50-53. You can simplify this sentence as too general;
Response: We have revised it.
Line 50-60. Introduce a more specific bibliography on viruses involved in the snap bean. From my analysis of the bibliography reported by you, there are no percentages of reduction due to snapbean viruses;
Response: We have revised it.
Line 86 Correlation or relationship?;
Response: We evaluated the correlations between the populations of insect and climatic factors.
Line95. What are energy trees?;
Response: We have revised it.
Lines 95-97. This sentence is superfluous. It is understandable that both trees and seasonal crops fall within the study area. I really don't understand the importance of this sentence. To simplify, you can insert a drawing with the distribution scheme of the parcels and neighboring areas;
Response: We have revised it. The rationale for including the information on neighboring areas was to consider the effects of different landscapes on insect populations.
Lines 100-103. Although this crop is autogamous sometimes visited by pollinating insects. It seems the value of pollination services to common bean yield is not known (see:10.7717/peerj.10102).
Response: Thank you for your information. Yes. We also include this information in the Discussion. In the Discussion, we stated “Although common beans are partially autogamous, several studies demonstrated that cross-pollination provided by insect pollinators could increase seed production by reducing abortion rates [34–36]”.
- Free, J.B. Insect pollination of crops, 2nd ed.; Academic Press: London, UK, 1993; pp. 684.
- Ibarr a-Perez, F.J.; Barnhart, D.; Ehdaie, B.; Knio, K.M.; Waines, J.G. Effects of insect tripping on seed yield of common beans. Crop Sci. 1999, 39, 428–433. https://doi.org/10.2135/crops ci1999.00111 83X00 39000 200022x.
- Ramos, D.D.L.; Bustamante, M.M.; Silva, F.D.D.S.E.; Carvalheiro, L.G. Crop fertilization affects pollination service provi-sion–common bean as a case study. PLoS One 2018, 13, e0204460. https://doi.org/10.1371/journal.pone.0204460.
Line 111. Specify the number of plants for each single experimental plot. It seems to be 2.7 m2, correct?;
Response: We have revised it. The row was 3.0 m in length and the row spacing was 0.9 m. There were four rows in each single experimental plot. So, the area of each plot was 8.1 m2 (0.9 m X 3 row-spacings X 3.0 m).
Lines 132-133. Specify the distance between the climatic station and the study area. Also are the experimental parcels and the climatic station in the same altimetry?
Response: We have revised it. Yes. The experimental parcels and the climatic station are in the same altimetry.
Line 151. Why did you choose to use more colors in the traps? Don't you think that for every siglolo experimental plot these traps are excessive as they may have conditioned the densities of some species?;
Response: The three colors (white, yellow, and blue) are the most common colors that used to trap pollinators (see “10.1093/aesa/sav057” and “10.3390/insects7040062”). We evenly distributed the pan traps in each single plot. In each experimental plot, each of the three pan traps (blue, yellow, and white) was randomly placed between the first and second rows, between the second and third rows, or between the third and fourth rows.
Line 162. Are you sure they're mostly pollinators?
Response: There are three common pollinators, such as bees, wasps, and moths. So, we regarded bees, wasps, and moths trapped in the pan traps as pollinators. For example, the pan traps also trapped many kudzu bugs. We regarded kudzu bugs as pests, not pollinators.
Line 173. Specify whether the choice of the sweeps time for each single experimental particle was causal;
Response: The choice of the sweeps time for each single experimental particle was not casual. We always started from the same plot. And samplings were initiated in the mornings (between 0800 hr and 1030 hr) to standardize evaluation. One person was responsible for one block. Each person can complete the sweep work in about one hour. So, we supposed that there was no significant differences in insect populations within one hour.
Line194. Each insect or each insect species?
Response: We want to say each insect including pest, pollinator, and natural enemy discovered in the experiment.
Line203. Are the authors with kudzu bug referring to Megacopta cribraria'?
Response: Yes. That is right. We have revised it.
Lines 295–296. Encarsia sp. and Eretmocerus sp.;
Response: We have revised it.
In paragraph 3.2.1. I don't understand why the results of a previous paper are reported;
Response: The rationale for including the previous data of B. tabaci was to present information of all the insects discovered in the field. All the insects included B. tabaci even we reported it previously. So, we only included a summary of data from B. tabaci. In the Conclusion, we summarized some cultivars which had the least numbers of both B. tabaci nymphs and L. lineolaris adults. So, the growers can use those cultivars with less B. tabaci nymphs and L. lineolaris adults.
Line422-423. Remove this sentence as it was specified in the previous lines.
Response: We double check this sentence of “The susceptibility of the above 24 local and commercially available snap bean cultivars to B. tabaci has been discussed in our previous study [13]”. We did not find that this sentence has been specified in the previous lines. Could you please clarify it?
Line454-456. I do not understand the need to justify through this reference as the mango. The species are very different and also in mango pollination is closely linked to pollinating insects.
Response: We have revised it.
Line 470-471. Nel riferimento bibliografico citato <<39>> non c’ènessun riferimento ad Empoasca fabae;
Response: We are not sure about the comments.
498-499. This statement "suppress B. tabaci, E. fabae, and L.lineolaris populations" seems excessive to me. A reduction in thepopulations of these pests is more likely.
Response: We have revised it.
Round 2
Reviewer 3 Report
Dear Authors,
I read carefully your new version of the mns Insects-2161923, you submitted after sharing my notes referred to the first draft. I appreciated your efforts into improving the quality presentation of the text. In particular, the drafting , in M&M section, of layout (Figure 1) and a more detailed and clear description of the experimental design. Moreover, some phrases added in Results and Discussion have improved the clearness of the comments recorded. The new version also cite 8 new references, all properly added.
My honest comment is that your contribution is ready to be considered to publish.
Sincerely

Reviewer 4 Report
The aforementioned MS has been revised according to the suggestions recommended in my review and so now I consider the MS improved in the different sections. However, I advise to report a table of abundances as suggested in the first phase of the review, in order to stimulate future comparisons with other similar studies.